# Association between Decreased ITGA7 Levels and Increased Muscle α-Synuclein in an MPTP-Induced Mouse Model of Parkinson’s Disease

**DOI:** 10.3390/ijms23105646

**Published:** 2022-05-18

**Authors:** Sangeun Han, Sabina Lim, Sujung Yeo

**Affiliations:** 1Department of Meridian and Acupoint, College of Korean Medicine, Kyung Hee University, Seoul 02453, Korea; sadgc0303@khu.ac.kr; 2Research Institute of Korean Medicine, Sang Ji University, Wonju 26339, Korea

**Keywords:** Parkinson's disease, ITGA7, alpha-synuclein, MPTP, muscle

## Abstract

Parkinson's disease (PD) is a neurodegenerative disease characterized by the loss of dopaminergic neurons in the substantia nigra (SN), reducing dopaminergic levels in the striatum and affecting motor control. Herein, we investigated the potential relationship between integrin α7 (*ITGA7*) and α-synuclein (α-syn) in the muscle of methyl-4-phenyl-1,2,3,6 tetrahydropyridine (MPTP)-induced mice and C2C12 cells. To characterize the pathology of PD, we examined the expression of tyrosine hydroxylase (TH) in the SN of the midbrain. Compared with the control group, MPTP-treated mice showed a significant decrease in TH expression in the SN, accompanied by a significant decrease in muscle ITGA7 expression. Compared with the control group, α-syn expression was increased in the MPTP group. Furthermore, the pattern of α-syn expression in the MPTP group was similar to the ITGA7 expression pattern in the control group (linear forms). To determine the relationship between ITGA7 and PD, we examined the expression of ITGA7 and α-syn after *ITGA7* knockdown using siRNA in C2C12 cells. ITGA7 expression significantly decreased while α-syn expression significantly increased in siRNA-treated C2C12 cells. These results suggest that decreased ITGA7 muscle expression could increase α-syn expression. Moreover, α-syn accumulation, induced by decreased muscle ITGA7, might contribute to PD pathology.

## 1. Introduction

Parkinson's disease (PD) is a neurodegenerative disease known to occur in 1–2 individuals per 1000 and is estimated to impact 1% of the population over 60 years of age [1]. The loss of dopaminergic neurons in the substantia nigra (SN) reduces dopamine levels in the striatum (ST) and affects motor control. PD is characterized by motor symptoms, such as bradykinesia, resting tremors, and muscle stiffness, and non-motor symptoms, such as sleep disturbance, depression, constipation, and loss of smell [2]. PD is caused by several environmental risk factors, with approximately 5–10% of cases attributed to genetic factors [3]. α-Synuclein (α-syn), associated with PD and other neurodegenerative diseases, is a presynaptic protein found to be elevated in PD and involved in dopaminergic cell death [4].

Integrin α7 *(ITGA7)* is a gene detected in the brain, muscle, and heart [5]. *ITGA7* is involved in various cell–cell and cell–extracellular matrix (ECM) interactions, such as cell growth, cell survival, and apoptosis [6]. Integrin alpha-7, a protein encoded by the gene *ITGA7*, belongs to the integrin alpha-chain family [7]. Integrin is a heterodimer-integrated membrane protein composed of alpha and beta chains, mediating cell–cell and cell–cytomatrix interactions, leading to cell migration, morphological development, differentiation, and metastasis [8]. ITGA7 is a laminin receptor in skeletal muscles, connecting the ECM to the internal actin cytoskeleton [9]. ITGA7 is highly expressed in cardiac, skeletal, and smooth muscle cells, and it has been reported that mutations in *ITGA7* can be associated with congenital myopathy and incompressible cardiomyopathy [10]. Furthermore, *ITGA7* reportedly plays a role in tumorigenesis and acts as a suppressor gene in breast cancer [11,12]

Notably, ITGA7, typically concentrated at the muscle–tendon junctions of postnatal muscles, is upregulated throughout the sarcolemma in Duchenne muscular dystrophy. Overexpression of ITGA7 prevents loss of force and increases the diaphragm-specific force following contraction-induced injury, supporting the application of ITGA7 as a potential therapeutic for Duchenne muscular dystrophy [13].

Given the association between ITGA7 and myopathy, ITGA7 might be potentially linked with the muscle in PD, considering the motor symptoms. In a previous study [14], which reduced ITGA7 expression can be associated with enhanced α-syn in the brain of MPTP-induced mice. In addition, we postulated that muscle ITGA7 expression is altered during PD. Accordingly, in the present study, we aimed to elucidate the mechanism of the action of ITGA7 and examined whether a decrease in muscle *ITGA7* expression increases α-syn during PD.

## 2. Results

### 2.1. Decreased Expression of Tyrosine Hydroxylase (TH) in SN and ST in an MPTP-Induced PD Mouse Model

Herein, control (CTL) and MPTP groups were intraperitoneally injected with saline and MPTP once daily. After 4 weeks, we analyzed changes in TH expression in the SN and ST to confirm the establishment of a chronic MPTP-induced PD mouse model. In both SN and ST, TH expression was significantly decreased in the MPTP group (Figure 1b,d) when compared with the CTL group (*p* < 0.005, *n* = 3, Figure 1a,c). In addition, TH reduction is more marked in SN and not in ST. MPTP induced neurodegeneration might affect more in SN than ST in our 4week MPTP mouse model. 

### 2.2. Behavior Tests in an MPTP-Induced PD Mouse Model

Behavior tests were performed to evaluate the motor abilities of mice in the CTL and MPTP groups. In the Rotatod test, the mice in the MPTP group fell earlier than the mice in the CTL group by an average of 44 *s* (*n* = 5, *p* < 0.005, Figure 1f). In the pole test, the mice in the MPTP group landed on the floor 0.61 s faster than the mice in the CTL group. Mice in the MPTP group seemed to lack leg strength to hold the pole and landed by sliding (*n* = 5, *p* < 0.05, Figure 1h).

### 2.3. Reduced Muscle Expression of ITGA7 in an MPTP-Induced PD Mouse Model

Using immunohistochemical analysis (IHC), we confirmed that muscle expression of ITGA7 was significantly reduced in the MPTP group (Figure 2c,d) when compared with the CTL group (Figure 2a,b). Similarly, Western blot analysis revealed that ITGA7 expression was decreased in the MPTP group when compared to that in the CTL group (*p* < 0.05, *n* = 3, Figure 2e,f).

### 2.4. Increased Muscle Expression of α-Syn in an MPTP-Induced PD Mouse Model

IHC confirmed that the muscle expression of α-syn was significantly higher in the MPTP group (Figure 3c,d) than in the CTL group (Figure 3a,b). Likewise, Western blot analysis showed that the expression of α-syn increased in the MPTP group when compared with that in the CTL group (*p* < 0.05, *n* = 3, Figure 3e,f).

### 2.5. Immunofluorescence Analysis of ITGA7 Co-Localized with α-Syn in Skeletal Muscle

Using immunofluorescence analysis, we observed that ITGA7 was co-localized with α-syn. In addition, *ITGA7* was more potently expressed in the CTL group than in the MPTP group (Figure 4b,g). In contrast, α-syn was more robustly expressed in the MPTP group than in the CTL group. Double-labeling of ITGA7 and α-syn revealed that the green fluorescence of ITGA7 was strongly expressed in the CTL group (Figure 4a,f), whereas α-syn expression was stronger in the MPTP group.

### 2.6. Western Blot Analysis of C2C12 Cells Transfected with ITGA7 siRNA

ITGA7 siRNA significantly decreased the expression of *ITGA7* in C2C12 cells (*p* < 0.005, *n* = 3, Figure 5a), whereas it significantly increased the expression of α-syn (*p* < 0.005, *n* = 3, Figure 5b).

### 2.7. ITGA7 Co-Localized with α-Syn in C2C12 Cells

Immunofluorescence analysis revealed the co-localization of ITGA7 and α-syn. Compared with the CTL group, the expression of ITGA7 was reduced (Figure 6g), whereas that of α-syn was enhanced (Figure 6f) in the MPP+ group. The CTL group showed combined expression of ITGA7 and α-syn (Figure 6c,d). The MPP+ group showed less combined expression of ITGA7 and α-syn and stronger α-syn expression compared to CTL group (Figure 6h,i).

## 3. Discussion

α-Syn is a small cytoplasmic protein present in presynaptic nerve terminals. Notably, it has been reported that α-syn misfolding and dysfunction can contribute to the pathogenesis of PD and neurodegenerative diseases [13]. 

Mutations in the ITGA7 gene can cause congenital myopathies, characterized by delayed developmental milestones and movement disorders, and the alpha7 subunit is known to be primarily involved in differentiation and migration processes during myogenesis [15,16]. In addition, ITGA7 interacts with dystrophin-related glycoproteins and is a muscle stem cell surface marker that influences myogenic progenitor cell generation and muscle differentiation [17]. 

Based on accumulated evidence [15,16,17], it can be suggested that ITGA7 deficiency could lead to impaired recovery of muscle cell generation and muscle differentiation and might further lead to muscle dysfunction. As motor dysfunction is the primary symptom of PD, we hypothesized that ITGA7 deficiency in muscles might be associated with PD pathology. Our results suggest that α-syn accumulation, induced by decreased muscle ITGA7, might be one of the underlying causes of PD. 

To confirm the establishment of the MPTP-induced PD mouse model, the expression of TH in the SN and ST was verified, which was found to be significantly reduced in the MPTP group when compared with the CTL group. Immunofluorescence analysis showed co-localization of α-syn and ITGA7 in the muscle, displaying decreased ITGA7 expression and increased α-syn expression in the MPTP group when compared with the CTL group. In addition, Western blot analysis confirmed that ITGA7 levels were reduced while α-syn levels were increased in the MPTP group. The linear expression pattern of ITGA7 in the CTL group was similar to that of α-syn in the MPTP group.

It has been reported that C2C12 cells express various types of integrins [18], and the ECM protein concentration on the culture surface can improve movement speed by modulating the behavior of C2C12 cells and cell adhesion on the culture surface via the regulation of integrin expression [19]. Herein, we confirmed that decreased ITGA7 expression induced by ITGA7 siRNA leads to increased α-syn expression in C2C12 cells. Elevated levels of α-syn protein in the muscle and other regions can enter the brain and induce the pathological process of PD [20]. Therefore, we suggest that increased α-syn levels induced by decreased ITGA7 levels may accelerate the pathological process of PD [21,22]. Consistent with a previous study examining changes in brain ITGA7 expression, the results of the present study also revealed that decreased ITGA7 expression could induce increased α-syn expression in the muscle. These results suggest that reduced ITGA7 expression in the brain and muscle could be an underlying cause of PD, accompanied by enhanced α-syn expression. In addition, given that ITGA7 and α-syn exhibit linear expression patterns, it can be speculated that α-syn may replace the decreased expression of ITGA7 in the SN region and muscle. Further studies are needed to determine whether α-syn can replace ITGA7.

## 4. Materials and Methods

### 4.1. MPTP Mouse Model 

In the present study, we used 4-week-old male inbred C57BL/6 mice (20–22 g; DBL, Daejeon, Korea) divided into CTL and MPTP groups. The CTL group was injected with 0.9% (100 μL) saline, and the MPTP group was intraperitoneally injected with MPTP-HCl (20 mg/kg of free base; Sigma, St. Louis, MO, USA) in 0.9% (100 μL) saline once daily for 4 weeks. The study protocol was approved by the Animal Care and Use Committee (IACUC) of Sangji University (approval code: 2018-5; date of approval 3 March 2018).

### 4.2. Behavior Tests

Rotarod and pole tests were performed to evaluate the motor ability of MPTP mice before the last MPTP injection. The rotarod treadmill diameter was 280 mm, and the test was performed in an accelerated mode for 4 min from 10 rpm to 50 rpm in 5-min running time. After 4 min of accelerated mode, 50 rpm was maintained for 1 min until completion. The time until the first fall or the first drop was measured. In the pole test, a wooden vertical pole (length 548 mm, diameter 8 mm) was used. The time the mouse took while moving from the top to the bottom of the pole was measured.

### 4.3. Cell Lines and Cultures 

Briefly, C2C12 cells were grown at 37 °C in humidified CO_2_ environment in Dulbecco’s modified Eagle’s medium (DMEM; GenDEPOT, Katy, TX, USA), supplemented with 10% fetal bovine serum (FBS; Lonza, Walkersville, MD, USA), and 100 U/mL of penicillin-streptomycin (Gibco, Amarillo, TX, USA).

### 4.4. ITGA7 Small Interference RNA

Stealth siRNA against *ITGA7* (NM_008398, 5′-GAC AUG CAC UAC CUC GUC U-3′) and negative control duplexes (i.e., scrambled siRNA against ITGA7, 5′-UUC UCC GAA CGU GUC ACG UTT-3′) were purchased from (Bioneer Inc., Daejeon, Korea). C2C12 cells were treated with ITGA7 siRNA for 24 h. Before siRNA transfection, C2C12 cells were incubated in Opti-MEM medium (Gibco, Amarillo, TX, USA). Transfection reagent was used in a 3.5:1 transfection reagent-to-duplex RNA ratio (Promega, Madison, WI, USA) in Opti-MEM medium.

### 4.5. MPP + Treatment

C2C12 cells were treated with 500 μM MPP + iodide (Sigma, St. Louis, MO, USA) for 18 h. MPP+ administration was performed simultaneously in the same medium 37 °C in humidified CO_2_ environment.

### 4.6. Western Blot Analysis 

C2C12 cells were homogenized using radioimmunoprecipitation assay buffer (RIPA) on ice for 20 min for Western blot analysis. The lysate was centrifuged at 12,000 rpm (4 °C for 20 min), and then the supernatant protein concentration was measured using BCA. A sample of 20 g total protein was separated by sodium dodecyl sulfate-polyacrylamide gel electrophoresis (SDS-PAGE; 4–15% Tris-Bis mini gel) and transferred to a PVDF membrane. Membranes were blocked with 3% bovine serum albumin at 37 °C for 90 min, followed by incubation with mouse anti-ITGA7 (1:1000; Santa Cruz Biotechnology, Santa Cruz, CA, USA; sc-515716) and anti-α -syn (1:500; Novus Biologicals, USA; NBP2-15365) antibodies overnight at 4 °C. After washing three times for 15 min with Tris-buffered saline (TBS) containing 0.01% Tween-20 (TBST), the membrane was treated with the secondary antibody for 1 h., followed by horseradish peroxidase (HRP)-conjugated anti-mouse or anti-rabbit IgG antibody (1:5000; Santa Cruz Biotechnology). After washing thrice for 15 min with TBST, protein bands were visualized using a chemiluminescent substrate. Muscle regions were homogenized using 20 mM RIPA on ice for 30 min. Then, the soluble supernatant (30 µg total protein) was centrifuged at 12,000 rpm for 20 min at 4°C. Antibodies and processes were identical to those for C2C12 cells. Membranes’ band density was determined using ImageJ (https://rsbweb.nih.gov/ij/, accessed on 1 January 2022) and normalized to that detected using anti-𝛽-actin antibody (1:2000; Santa Cruz Biotechnology; sc-47778).

### 4.7. Immunohistochemistry 

Briefly, the brain and muscles of mice were fixed in 0.05 M sodium phosphate buffer containing 4% paraformaldehyde at 4 °C for 24 h, followed by dehydration with sucrose at 4 °C for 2 days, and then freeze-dried. Each section was cut using a cryomicrotome (40 μm thickness). Subsequently, muscle samples were incubated with anti-integrin alpha7 (1:100; Santa Cruz Biotechnology) and anti-α-syn antibody (1:500; Novus Biologicals), while brain tissue samples were incubated with anti-TH antibody (1:2000; Santa Cruz Biotechnology; sc-25269) for 24 h at 4 °C. Each section was then treated with biotinylated anti-mouse IgG, avidin-biotin-peroxidase complex, and diaminobenzidine hydrogen peroxide solution.

### 4.8. Immunofluorescence 

After incubation with primary antibodies and then with biotinylated anti-mouse IgG, each section was treated with fluorescein avidin DCS (Vector Laboratories, Burlington, ON, CA) for ITGA7 (1:100) and α-syn (1:200). Subsequently, sections were treated with an avidin/biotin blocking kit and a M.O.M mouse Ig blocking reagent (Vector Laboratories, Burlingame, CA, USA), followed by staining with anti-α-syn or anti-Itga7 IgG at 4 °C overnight. Each section was treated with biotinylated anti-mouse IgG, followed by incubation with rhodamine avidin D (Vector Laboratories, Burlingame, CA, USA). Images were obtained using a Nikon X-cite series 120 Q microscope (Nikon, Tokyo, Japan), and the exposure parameters were the same for each group of samples.

### 4.9. Statistical Analysis

Data analyses were performed using Student’s *t*-test with SPSS 25 (IBM Corp., Armonk, NY, USA). Data values are presented as mean ± standard error.

## 5. Conclusions

In conclusion, decreased ITGA7 expression and increased α-syn expression were observed in the muscle of an MPTP-induced PD mouse model. Furthermore, we confirmed that decreased ITGA7 expression induced by ITGA7 siRNA resulted in increased α-syn expression in C2C12 cells. These results suggest that α-syn accumulation induced by a decrease in ITGA7 in the muscle might be an underlying cause of PD pathology. 

## Figures and Tables

**Figure 1 ijms-23-05646-f001:**
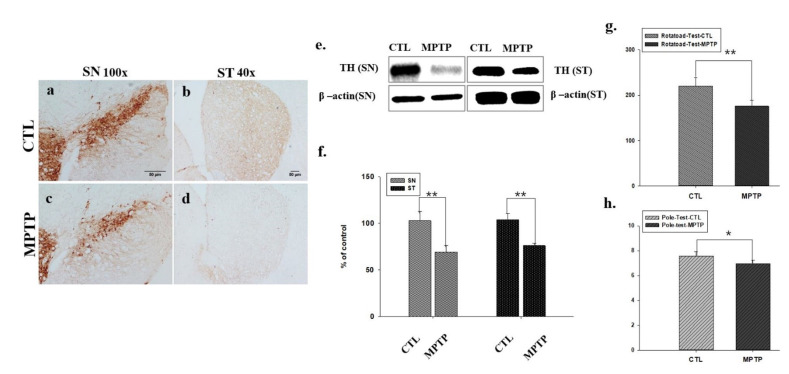
Immunohistochemistry of tyrosine hydroxylase (TH) expression in substantia nigra (SN, **a**,**c**) and striatum (ST, **b**,**d**) of control (CTL) and 1-methyl-4-phenyl-1,2,3 Chemical analysis, 6-tetrahydropyridine treatment (MPTP) group. TH expression was decreased in the SN and ST regions of the brain of MPTP-treated mice compared to CTL. Similarly, to Western blot analysis (**e**), the expression of TH was significantly reduced in the MPTP group compared to the CTL group (**f**). In the rotarod (**g**) and pole tests (**h**), when compared with the CTL group, the MPTP group showed a significant decrease in motor function. * *p* < 0.05, ** *p* < 0.005 compared to CTL. All values are expressed as mean standard error, and statistical analysis was performed using Student’s *t*-test.

**Figure 2 ijms-23-05646-f002:**
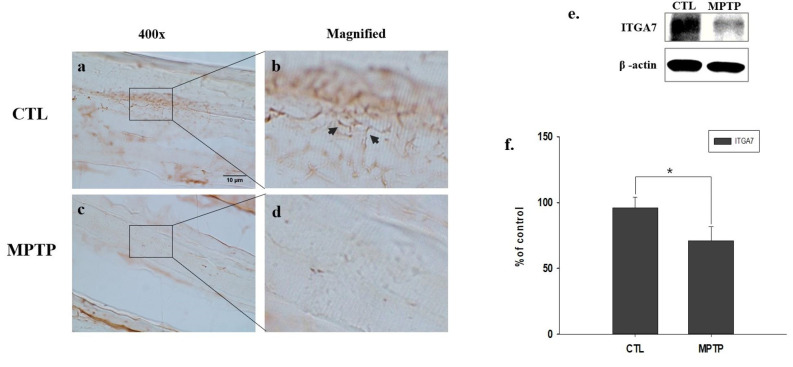
Representative images of muscle integrin alpha-7 (ITGA7) expression in a chronic 1-methyl-4-phenyl-1,2,3,6-tetrahydropyridine (MPTP)-induced Parkinson’s disease mouse model. ITGA7 expression increases in the control (CTL) group (**a**,**b**) and decreases in the MPTP group (c, d). Images of (**b**) and (**d**) are magnified images of squares in (**a**) and (**c**). Western immunoblot analyses (**e**,**f**) show that ITGA7 expression is decreased in the MPTP group. * *p* < 0.05 compared to CTL. All values are expressed as mean ± standard error, and statistical analyses were performed using the Student’s *t*-test. Black arrows indicate regions of interest.

**Figure 3 ijms-23-05646-f003:**
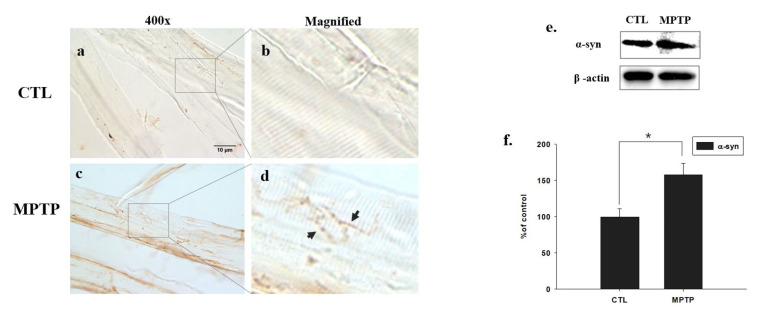
Representative images of alpha-synuclein (α-syn) expression in the muscle of chronic 1-methyl-4-phenyl-1,2,3,6-tetrahydropyridine (MPTP)-induced Parkinson’s disease mouse model. α-syn expression decreases in the control (CTL) group (**a**,**b**) and increases in the MPTP group (**c**,**d**). Images of (**b**) and (**d**) are magnified images of squares in (**a**) and (**c**). Western immunoblot analyses (**e**,**f**) show that α-syn expression increases in the MPTP group. * *p* < 0.05 compared to CTL. All values are expressed as mean ± standard error, and statistical analyses were performed using the Student’s *t*-test. Black arrows indicate regions of interest.

**Figure 4 ijms-23-05646-f004:**
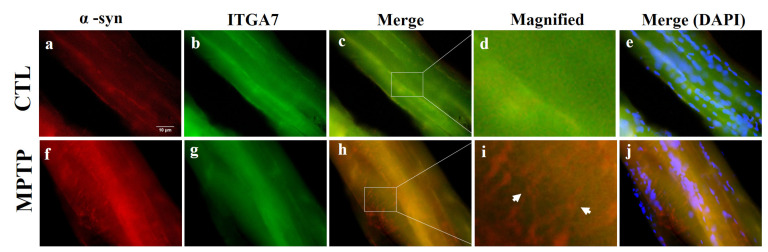
Immunofluorescence images of integrin alpha-7 (ITGA7) co-localized with alpha-synuclein (α-syn) in the muscle of a 1-methyl-4-phenyl-1,2,3,6-tetrahydropyridine (MPTP)-induced Parkinson’s disease mouse model. Muscle regions were immunofluorescently labeled with anti-ITGA7 (**b**,**g**) and anti-α-syn (**a**,**f**) antibodies using Rhodamine Avidin (**a**,**f**; red), then double immunolabeled with ITGA7 antibodies using Fluorescein Avidin (**b**,**g**; green). The ‘Merge’ panels (**c**,**h**) show merged images of the individual α-syn (**a**,**f**) and ITGA7 (**b**,**g**) panels. In the control group (**c**), considerably stronger ITGA7 green fluorescence can be observed. In the MPTP group (**h**), intense red fluorescence can be observed. Images of (**d**) and (**i**) show enlarged images of square in (**c**) and (**h**). (**e**,**j**) show merged images with DAPI. White arrows indicate regions of interest.

**Figure 5 ijms-23-05646-f005:**
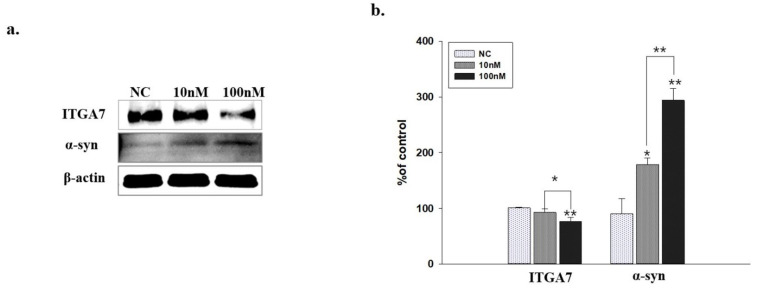
Western immunoblot analysis shows that administration of ITGA7 siRNA reduced *ITGA7* expression in C2C12 cells (**a**). It also shows that α-synuclein (*α-syn*) expression was increased (**b**). ITGA7 siRNA treatment (10 nM for 24 h); ITGA7 siRNA treatment (100 nM for 24 h). * Negative control (NC) siRNA treatment (100 nM for 24 h). * *p*< 0.05, ** *p*< 0.005 compared to NC. All values were expressed as mean ± standard error, and statistical analysis was performed using Student’s *t*-test.

**Figure 6 ijms-23-05646-f006:**
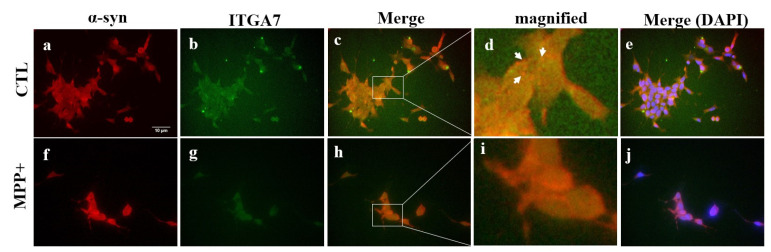
Immunofluorescence images of integrin alpha-7 (ITGA7) co-localized with alpha-synuclein (α-syn) in C2C12 cells. Cell regions were immunofluorescently labeled with anti-ITGA7 (**b**,**g**) and anti-α-syn (**a**,**f**) antibodies using Rhodamine Avidin (**a**,**f**; red), followed by Fluorescein Avidin (**b**,**g**; green). Middle panels (**c**,**h**) show merged images of individual α-syn (**a**,**f**) and ITGA7 (**b**,**g**) panels. Compared with the control (CTL) group, the expression of ITGA7 decreases and the expression of α-syn increases in the MPP+ group (**h**,**i**), and merged expression of ITGA7 and α-syn can be observed in the CTL group (**c**,**d**). (**d**) and (**i**) are enlarged images of the squares in (**c**) and (**h**). (**e**,**j**) displays the image merged with DAPI. White arrows indicate regions of interest.

## Data Availability

All data analyzed in this study are included in this published article.

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
