# Peer review of "Association between Decreased ITGA7 Levels and Increased Muscle α-Synuclein in an MPTP-Induced Mouse Model of Parkinson’s Disease"

_ijms, 2022, doi:10.3390/ijms23105646_

Round 1

Reviewer 1 Report

The work describes the strict correlation between ITGA7 and α-synuclein levels in the muscle in a MPTP-induced mouse model of Parkinson disease. Using siRNA technique in C2C12 cell line, the authors confirm the strict correlation between α-synuclein accumulation and ITGA7 decreased expression.

The work is interesting and relevant for the field: in fact, I think it is can be useful for further progress in understanding Parkinson disease etiology and characteristics, not only at central level, but also at peripheral one.

However, I have some important questions/considerations the authors would like to answer/comment:

  • The first important consideration is that the number of animals per group is not reported, together with the number of samples considered for Western blot and immunohistochemical analysis.
  • Another important point to clarify concerns western blot images: in fig.1 for SN and ST the same β actin is reported, but SN and ST are two different tissues. Did the authors forget to add the other β actin?

Moreover, β actin in fig 3 should be different from one showed in fig.2, since, according to original blot file, lanes 3 and 4 are considered for α syn, while lanes 1 and 2 for ITGA7.

Finally, according to original blot file, in fig 5 β actin for ITGA7 lacks: it can be the same for α syn since are considered different lanes for α syn and ITGA7.

Then, some minor points to clarify:

  • At line 64 remove the bracket after SN;
  • Invert the words “MPTP” and “saline” at line 66-67;
  • In par. 2.1. fig 1e and 1f are not analyzed. Please also comment why TH reduction is more marked in SN and not in ST.
  • In order to better understand figure 1, please add to which tissue the immunohistochemistry images refer (SN or ST).
  • Please, correct “d” and “e” with “c” and “d” at line 86-87.
  • Please better explain Figure 4 (par.2.4) and its legend. It is not clear for this reviewer.
  • Please replace “MPTP+” at line 133 and line 141 with “MPP+”;
  • In par.2.6. the fig.6 is partially commented.
  • In discussion section, the authors wrote: “Based on accumulated evidence, it can be suggested that ITGA7 deficiency could lead to impaired recovery of muscle cell generation and muscle differentiation and might further lead to muscle dysfunction” (lines 154-156). Add a reference.

The sentence at lines 166-167 is not clear. Please, reformulate.

The authors wrote:” Therefore, increased α-syn levels induced by decreased 174 ITGA7 levels may accelerate the pathological process of PD” (lines 174-175). Please, add a reference.

  • In Materials and Methods sections, please add where MPTP was purchased. Please also add for all antibodies used for western blot, immunohistochemistry and immunofluorescence where were purchased from and their catalogue number.
  • Please, add treatment conditions for in vitro experiments (par. 4.4), for example, which medium MPP+ was incubated in?
  • At line 216 probably “primary” should be replaced with “secondary”.

Reviewer 2 Report

   This paper has scientific merits in trying to connect decreased ITGA7 level and increased α-synuclein in PD. In previous study, the authors have shown that reduced ITGA7 level can enhance α-synuclein in the brain of MPTP-induced PD mice, in this manuscript, they focused on the relationship between muscle ITGA7 and α-syn levels in MPTP-induced mice. They found that ITGA7 expression significantly decreased while α-syn expression significantly increased in MPTP-induced PD mice and ITGA7 siRNA-treated C2C12 cells. The authors concluded that α-syn accumulation induced by decreased muscle ITGA7 may contribute to PD pathology.

There are a couple of minor points:

  1. In line 57-59, references related to reduced ITGA7 expression can be associated with enhanced α-syn in the brain of MPTP-induced mice should be listed.
  2. 1f and Fig.5b, statistical graphs should be adjusted accurately.
  3. For all the figures in this article should be mentioned in ‘Results’ and ‘Figure legends’.
  4. 1, the authors need to do some behavior tests including open field test, pole test, rotarod test to confirm the PD mouse model.
  5. 2b, Fig.3d, Fig.4i, Fig.6d, what the arrows represent should be mentioned.
  6. In line 119-120, the paragraph is not formatted correctly.
  7. 2-4, does ‘muscle’ refer to ‘Skeletal muscle’.

Round 2

Reviewer 1 Report

The authors replied to my answers/comments in an exhaustive manner.